# An exact chiral amorphous spin liquid

G. Cassella [1], P. d'Ornellas[1] ✉, T. Hodson[1], W. M. H. Natori [2] & J. Knolle [1,3,4] ✉

Topological insulator phases of non-interacting particles have been generalized from periodic crystals to amorphous lattices, which raises the question whether topologically ordered quantum many-body phases may similarly exist in amorphous systems? Here we construct a soluble chiral amorphous quantum spin liquid by extending the Kitaev honeycomb model to random lattices with fixed coordination number three. The model retains its exact solubility but the presence of plaquettes with an odd number of sides leads to a spontaneous breaking of time reversal symmetry. We unearth a rich phase diagram displaying Abelian as well as a non-Abelian quantum spin liquid phases with a remarkably simple ground state flux pattern. Furthermore, we show that the system undergoes a finite-temperature phase transition to a conducting thermal metal state and discuss possible experimental realisations.

Amorphous materials are condensed matter systems characterised by short-range regularities, and an absence of long-range crystalline order as studied early on for amorphous semiconductors[1,2]. The bonds of a wide range of covalent compounds can enforce local constraints around each ion, e.g. a fixed coordination number $z$, which has enabled the prediction of energy gaps even in lattices without translational symmetry[3,4], the most famous example being amorphous Ge and Si with $z = 4$[5,6]. Recently, following the discovery of topological insulators (TIs), it has been shown that similar phases can exist in amorphous systems characterised by protected edge states and topological bulk invariants[7–13]. However, research on electronic systems has been mostly focused on non-interacting systems with few exceptions, for example, to account for the observation of superconductivity[14–18] in amorphous materials or very recently to understand the effect of strong electron repulsion in TIs[19].

Magnetism in amorphous systems has been investigated since the 1960s, mostly through the adaptation of theoretical tools developed for disordered systems[20–23] and with numerical methods[24,25]. Research has focused on classical Heisenberg and Ising models, which are able to describe ferromagnetic, disordered antiferromagnetic and widely observed spin glass behaviour[26]. However, the role of spin-anisotropic interactions and quantum effects in amorphous magnets has not been addressed. It is an open question whether frustrated magnetic interactions on amorphous lattices can give rise to genuine quantum phases, i.e. to long-range entangled quantum spin liquids (QSL)[27–30].

The combination of a fixed local coordination number in conjunction with magnetic frustration generated by bond-anisotropic

Ising exchanges can lead to stable QSL phases. The seminal Kitaev model on the honeycomb lattice[31] provides an exactly solvable model whose ground state is a QSL characterised by a static $\mathbb{Z}_2$ gauge field and Majorana fermion excitations. Several instances of Kitaev candidate materials have been synthesised in the last decade[32–36] following the suggestion that heavy-ion Mott insulators formed by edge-sharing octahedra may realise dominant Kitaev interactions[32]. In particular, recently it has been shown that the Kitaev material $Li_2IrO_3$ can be created with an amorphous structure[37]. In fact, with sufficiently fast cooling, any crystalline material can be made amorphous[2,38], opening the possibility for exploring a wide variety of non-crystalline Kitaev materials.

It is by now well known that the Kitaev model on any three-coordinated ($z = 3$) graph has conserved plaquette operators and local symmetries[39,40] which allow for a mapping onto an effective free Majorana fermion problem in a background of static $\mathbb{Z}_2$ fluxes[41–44]. However, in general this neither means that any $z = 3$ lattice Kitaev model can be straightforwardly constructed, nor that the QSL properties are obvious. Several obstacles remain. First, the labelling of bonds necessary to create a soluble Hamiltonian can be an NP-complete problem. Second, once the Majorana system has been constructed, determining the ground state out of the exponentially large number of $\mathbb{Z}_2$ flux sectors is generically hard, since Lieb's theorem – which defines the ground state flux configuration for the honeycomb – is not applicable for most lattices. Previous studies have relied on translation and reflection symmetries to reduce the number of sectors that must be checked[43,45,46], which cannot be done in an amorphous

[1]Blackett Laboratory, Imperial College London, London SW7 2AZ, United Kingdom. [2]Institut Laue-Langevin, BP 156, 41 Avenue des Martyrs, 38042 Grenoble Cedex 9, France. [3]Department of Physics TQM, Technische Universität München, James-Franck-Straße 1, D-85748 Garching, Germany. [4]Munich Center for Quantum Science and Technology (MCQST), 80799 Munich, Germany. ✉e-mail: ppd.dorn@gmail.com; j.knolle@tum.de

system. Third, once the ground state flux sector is found, it needs to be determined whether lattice disorder induces a gapless phase[47–49] or whether the fermionic spectrum is gapped, possibly with non-trivial topology[43].

In this article we study the Kitaev model on amorphous lattices and establish it as an example of a topologically ordered amorphous QSL phase. Concentrating on random networks generated via Voronoi tessellation[7,9] with $z = 3$, we show how to colour the bonds consistently. We find that the presence of plaquettes with an odd-number of sites lead to a chiral QSL with spontaneously broken time-reversal symmetry (TRS)[43,50–55]. We establish via extensive numerics that the ground state $\mathbb{Z}_2$ flux sector follows a remarkably simple counting rule consistent with Lieb's theorem[56]. We map out the phase diagram of the model and show that the chiral phase around the symmetric point is gapped and characterised by a quantised local Chern number $\nu$[7,57] as well as protected chiral Majorana edge modes. Finally, we discuss the effect of additional bond disorder and comment on the role of finite temperature fluctuations, showing that the proliferation of flux excitations leads to an Anderson transition to a thermal metal phase[47–49].

## Methods

We start with a brief review of the Kitaev model on the honeycomb lattice[31] before generalising it to amorphous systems. A spin-1/2 is placed on every vertex and each bond is labelled by an index $\alpha \in \{x, y, z\}$. The bonds are arranged such that each vertex connects to exactly one bond of each type. The Hamiltonian is given by

$$\mathcal{H} = -\sum_{\langle j,k \rangle_\alpha} J^\alpha \sigma_j^\alpha \sigma_k^\alpha, \tag{5}$$

where $\sigma_j^\alpha$ is a Pauli matrix acting on site $j$, $\langle j, k \rangle_\alpha$ is a pair of nearest-neighbour indices connected by an $\alpha$-bond with exchange coupling $J^\alpha$. For each plaquette of the lattice, we define a flux operator $W_p = \prod \sigma_j^\alpha \sigma_k^\alpha$, where the product runs clockwise over the bonds around the plaquette. These commute with one another and the Hamiltonian, so correspond to an extensive number of conserved quantities. This allows us to split the Hilbert space according to the eigenvalues $\phi_p = \pm 1$ ($\pm i$ for odd plaquettes) of $\{W_p\}$.

The Hamiltonian in Eq. (5) can be exactly solved by transforming to a Majorana fermion representation[31], see Fig. 1. Each spin is represented with four Majorana operators, $\sigma_i^\alpha = ib_i^\alpha c_i$. We define a set of conserved bond operators $\hat{u}_{jk} = ib_j^\alpha b_k^\alpha$. As with the $W_p$ operators, we may partition the Majorana Hilbert space according to the eigenvalues of these operators, $u_{jk} = \pm 1$. For a given choice of these bond variables, Eq. (5) reduces to a quadratic Majorana Hamiltonian

$$\mathcal{H} = \frac{i}{4} \sum_{j,k} A_{jk} c_j c_k, \tag{6}$$

where $A_{jk} = 2J^\alpha u_{jk}$.

The matrix $iA$ determines properties of the fermionic degrees of freedom for a given flux configuration $\{u_{jk}\}$. The spectrum is obtained by rotating to a new Majorana basis consisting of pairs of operators $\tilde{c}_j', \tilde{c}_j''$, defined by a matrix $\tilde{c}_j = R_{jk} c_k$ containing the fermionic eigenstates. The Hamiltonian takes the form $\mathcal{H} = \sum_j \varepsilon_j i \tilde{c}_j' \tilde{c}_j''$, and in what follows we refer to fermionic properties of the system as those determined by $iA$ in a fixed flux sector.

The Kitaev Hamiltonian remains exactly solvable on any lattice in which no site connects to more than one bond of the same type[41]. Thus, we shall restrict our investigation to lattices in which every vertex has coordination number $z \leq 3$. Here we generate such lattices using Voronoi tessellation[58]. Once a lattice has been generated, the bonds must be labelled in such a way that no vertex touches multiple edges of the same type, which we refer to as a three-edge colouring. The problem of finding such a colouring is equivalent to the classical

problem of four-colouring the faces, which is always solvable on a planar graph[59,60] but can take up to seven colours on the torus[61]. In practice, we reduce the colouring to a Boolean satisfiability problem[62] with details described in the Supplementary Material. One example of a coloured amorphous lattice is shown in Fig. 1a.

Once the lattice and colouring has been found, the amorphous Hamiltonian is diagonalised using the same procedure as for the honeycomb model. Note that the Majorana system is only strictly equivalent to the initial spin system after a parity projection[63,64], details of which for the amorphous case are described in the Supplementary Material. Nevertheless, one can still use Eq. (6) to evaluate the expectation values of operators that conserve $\hat{u}_{jk}$ in the thermodynamic limit[65,66]. The ground state energy of a given flux sector is the sum of the negative eigenvalues of $iA/4$ in Eq. (6), and excitation energies are given by the positive eigenvalues.

## Results

We first investigate which flux patterns minimise the ground state energy on the amorphous lattice. When represented in the Majorana Hilbert space, flux operators $W_p = \prod \sigma_j^\alpha \sigma_k^\alpha$ correspond to ordered products of link variables $\hat{u}_{jk}$, and their eigenvalues describe the $\mathbb{Z}_2$ flux through each plaquette,

$$\phi_p = \prod_{(j,k) \in \partial p} -iu_{jk}, \tag{1}$$

where the product is taken over the $u_{jk}$ values going clockwise around the border $\partial p$ of each plaquette. We refer to a particular choice of a set of $\{\phi_p\}$ as a flux sector.

The spin Hamiltonian is real, thus it has TRS. However, the flux $\phi_p$ through any plaquette with an odd number of sides has imaginary eigenvalues $\pm i$. Thus, states with a fixed flux sector spontaneously break TRS, which in the context of crystalline Kitaev models was first described by Yao and Kivelson[67]. All flux sectors come in degenerate pairs, where time reversal is equivalent to inverting the flux through every odd plaquette[43,46].

For a system with $n_p$ plaquettes in periodic boundaries, there are $2^{n_p-1}$ possible flux sectors, and in general it is a nontrivial task to determine which pair of flux sectors has the lowest energy. On the honeycomb lattice, the ground state was shown by Lieb to be flux free, $\phi_p = +1$[56], however no such proof exists for amorphous lattices, since all lattice symmetries are broken.

To numerically determine the ground state flux sector, we first test a large number of finite size lattices (~25,000 lattices with 16 plaquettes), directly enumerating all possible flux configurations to find the lowest energy. In practice, care must be taken to account for finite size effects, as well as to ensure that the results hold as system size is increased – detailed in the Supplementary Material. Remarkably, we find that the energy is always minimised by setting the flux through each plaquette $p$ to

$$\phi_p^{\text{g.s.}} = -(\pm i)^{n_{\text{sides}}}, \tag{2}$$

where $n_{\text{sides}}$ is the number of edges that form the plaquette and the global choice of the sign of $i$ gives the two TRS-degenerate ground state flux sectors. The conjecture is consistent with results found on other regular lattices for which Lieb's theorem is not applicable[42]. Having identified the ground state, any other flux sector can be characterised by the configuration of vortices, i.e. by the plaquettes whose flux is flipped with respect to $\{\phi_p^{\text{g.s.}}\}$.

The ground state phase diagram can then be determined by varying the strength of each bond type, $J^\alpha$ while remaining in the ground state flux sector, and we numerically calculate the ternary phase diagram shown in Fig. 1c. The diagram contains two distinct phases: close to the corners of the triangle, e.g. $|J^z| \gg |J^x|, |J^y|$, the (A)

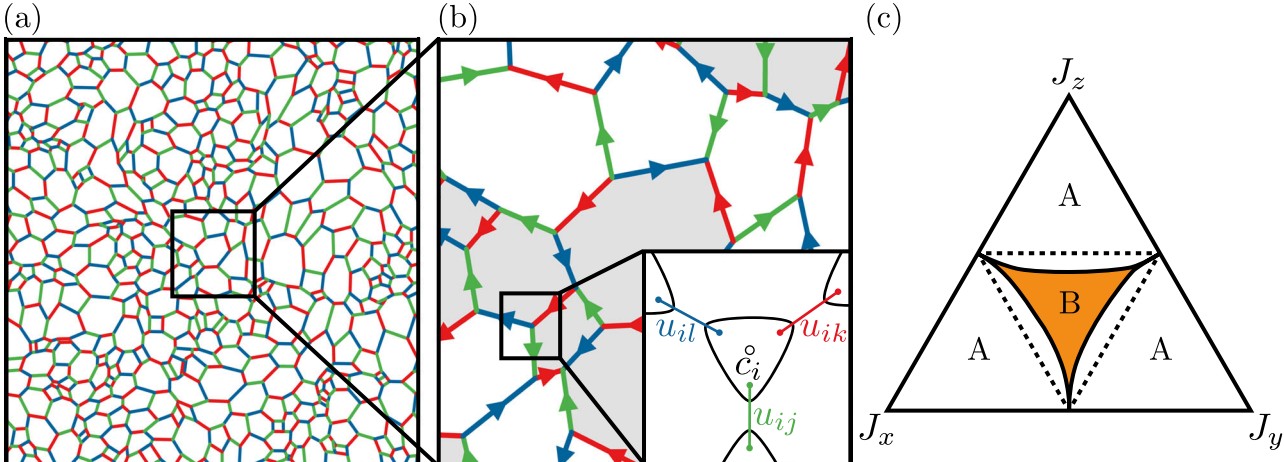

**Fig. 1 | Construction details for the amorphous lattice model. a** Amorphous lattice generated via Voronoi tessellation of a uniformly distributed random point set on the unit square. Periodic boundary conditions are imposed by tiling the unit square before Voronoi tessellation. **b** Magnified portion of the amorphous lattice. Arrows from site $j$ to site $k$ indicate direction where the bond variable $u_{jk} = 1$. An arbitrary flux sector is shown, where shaded plaquettes have $\mathbb{Z}_2$ flux flipped with respect to the ground state. Colours correspond to a valid assignment of the bond colourings, $\alpha_{jk}$. The inset demonstrates the Majorana construction on a tri-coordinate motif, which allows for the exact solution of the model. **c** Ternary phase diagram of the amorphous Kitaev model with varying exchange coupling. The isotropic regime $|J_x| \approx |J_y| \approx |J_z|$ (B), exhibits a topologically non-trivial chiral QSL ground state with Chern number $\nu = \pm 1$. The fermion gap of the ground state flux sector closes at the phase boundary (solid black lines), and a transition occurs to a $\nu = 0$ phase (A) for anisotropic couplings. The phase boundary was obtained by averaging over 20 amorphous lattice realisations with ~400 sites. Dotted black lines indicate the corresponding phase boundaries in the honeycomb model.

phase is equivalent to the toric code on an amorphous lattice[68]. The phase has a fermionic gap and supports Abelian excitations. Around the isotropic point $J^x = J^y = J^z$, the (B) phase is also gapped in contrast to the honeycomb case as a consequence of TRS breaking from the finite density of odd plaquettes. All lattices studied in this work were generated from a voronoi lattice with completely random seed points, and so had on average equal proportions of odd and even plaquettes. We will confirm below that the (B) phase is indeed a *chiral spin liquid*.

As the values of $J^\alpha$ are varied, the fermionic gap closes at the boundary between the two phases. In the honeycomb model, the phase boundaries are located on the straight lines $|J^\alpha| = |J^\beta| + |J^\gamma|$, for any permutation of $\alpha, \beta, \gamma \in \{x, y, z\}$. We find that on the amorphous lattice these boundaries exhibit an inward curvature similar to honeycomb Kitaev models with flux[69] or bond disorder.

Note, the presence of the gapped B-phase is non-trivial and related to our choice of homogeneous couplings for each colour of the bonds. In the Supplementary Material we study the robustness of the B phase with respect to bond disorder, e.g. a bond-length dependence of the interaction strength. In general one might expect disorder to lead to a gap closing, however we find that the gap is reduced but remains robust up to sizeable bond disorder.

A fundamental tool for understanding the distinction between the two phases is the Chern number. The original definition relies on momentum space, and so cannot be used here, where the system lacks any translational symmetry. However, recently methods have been developed for evaluating a real-space analogue of the Chern number[70,71]. Here we shall use a slight modification of Kitaev's definition[7,31,57]. For a choice of flux sector, we calculate the projector $P$ onto the negative energy eigenstates of the matrix $iA$ defined in Eq. (6). The local Chern number around a point $\mathbf{R}$ in the bulk is given by

$$\nu(\mathbf{R}) = 4\pi \mathrm{Im}\ \mathrm{Tr}_{\mathrm{Bulk}}\left(P\theta_{R_x}P\theta_{R_y}P\right), \qquad (3)$$

where $\theta_{R_x}$ is a step function in the $x$-direction, with the step located at $x = R_x$, $\theta_{R_y}$ is defined analogously. The trace is taken over a region around $\mathbf{R}$ in the bulk of the material, where care must be taken not to include any points close to the edges. Provided that the point $\mathbf{R}$ is sufficiently far from the edges, this quantity will be very close to quantised to the Chern number.

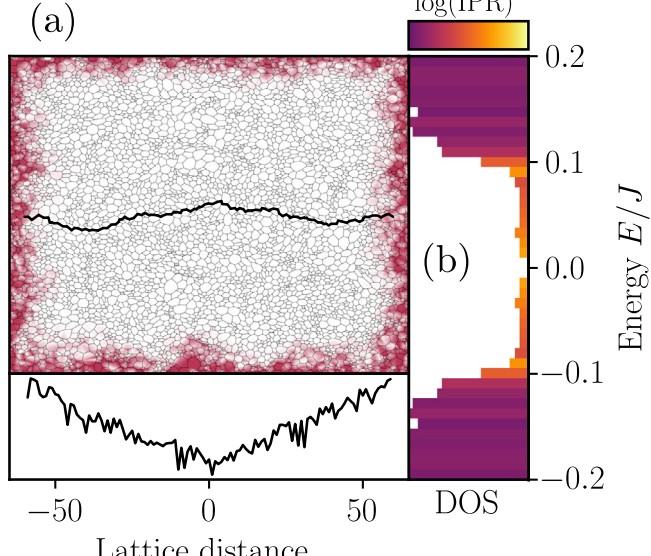

**Fig. 2 | Ground-state flux sector wavefunctions and spectrum. a** In-gap fermionic wavefunction drawn from the ground state flux sector in open boundary conditions. Colour indicates the local number density, showing a topological edge mode. The number density for this state along a line of lattice sites spanning the system (black line) is shown in the bottom subfigure on a logarithmic scale, demonstrating the characteristic exponential decay of topological edge modes with distance from the edge. **b** Ground-state flux sector fermionic density of states in open boundary conditions, coloured by inverse participation ratio. The increased inverse participation ratio of the in-gap states signifies their localisation to the edges of the system.

Using this local Chern marker, we determine that the (A) phase has Chern number $\nu = 0$, whereas the two TRS-degenerate ground state flux sectors in the (B) phase have Chern number $\nu = \pm 1$ respectively. In closed boundaries, this leads to the appearance of gap-crossing protected edge modes, in accordance with the bulk-boundary correspondence[72], an example is shown in Fig. 2. The edge modes are exponentially localised to the boundary of the system, and can be

qualitatively distinguished from bulk states by their large inverse participation ratio,

$$\text{IPR} = \int d^2 r |\psi(\mathbf{r})|^4, \qquad (4)$$

where $\psi(\mathbf{r})$ denotes an eigenmode of the free Majorana Hamiltonian derived in Eq. (6). Finally, we note that the closing of the fermionic gap on the boundary between the two phases is necessary in order to transition between states with different Chern numbers.

**Anderson transition to a thermal metal**
Having understood the spontaneous formation of a chiral amorphous QSL ground state, we are now in a position to discuss the finite temperature behaviour of the model. In general, an Ising-like thermal phase transition into the chiral QSL phase is expected akin to the one observed for the Yao–Kivelson model[73] but a full Monte-Carlo sampling, which is further complicated by the inherent disorder in the amorphous lattice, is beyond the scope of this letter. Nevertheless, the main effect of increasing temperature is the proliferation of fluxes which allow us to gain a qualitative understanding of the finite temperature behaviour[69].

On the honeycomb Kitaev model with explicit TRS breaking, Majorana zero modes bind to fluxes forming Ising non-Abelian anyons[74]. Their pairwise interaction decays exponentially with separation[48,49,75]. As temperature is increased, the proliferation of vortices in the system produces a finite density of anyons and their hybridisation leads to an Anderson transition to a macroscopically degenerate state known as a *thermal metal phase*[47,48,76]. This exotic phase has two key signatures. Firstly, the metallic phase is defined by a closing of the fermion gap – that is, it is driven by vortex configurations with a gapless fermionic spectrum. Secondly, we expect the density of states in a thermal metal to diverge logarithmically with energy and display characteristic low energy oscillations predicted by random matrix theory[47,77]. In the Supplementary Material we present numerical evidence showing that all of the above features carry over to the amorphous QSL with spontaneous TRS breaking, giving strong evidence for the transition to the thermal metal phase.

## Discussion
We have studied an extension of the Kitaev honeycomb model to amorphous lattices with coordination number $z = 3$. We found that it is able to support two quantum spin liquid phases that can be distinguished using a real-space generalisation of the Chern number. The presence of odd-sided plaquettes results in a spontaneous breaking of TRS, and the emergence of a chiral spin liquid phase. Furthermore we found evidence that the amorphous system undergoes an Anderson transition to a thermal metal phase, driven by the proliferation of vortices with increasing temperature. Our exactly soluble chiral QSL provides a first example of a topologically quantum many-body phase in amorphous magnets, which raises a number of questions for future research.

First, a numerically challenging task would be a study of the full finite temperature phase diagram via Monte-Carlo sampling and possible violations of the Harris criterion for the Ising transition stemming from the inherent lattice disorder[78–80]. Second, it would be worthwhile to search for experimental realisations of amorphous Kitaev materials, which can possibly be created from crystalline ones using standards method of repeated liquifying and fast cooling cycles[3,21,23]. The putative QSL behaviour of the intercalated Kitaev compound $H_3LiIr_2O_6$[81,82] could possibly be related to amorphous lattice disorder. Moreover, metal organic frameworks are promising platforms forming amorphous lattices[83] with recent proposals for realising strong Kitaev interactions[84] as well as reports of QSL behaviour[85]. We expect that an experimental signature of a chiral amorphous QSL is a half-quantised thermal Hall effect similar to magnetic field induced behaviour of honeycomb Kitaev materials[86–89]. Alternatively, it could be characterised by local probes such as spin-polarised STM[90–92] and the thermal metal phase displays characteristic longitudinal heat transport signatures[74]. Third, it would be interesting to study the stability of the chiral amorphous Kitaev QSL with respect to perturbations[93–97] and, importantly, to investigate whether QSL may exist for spin-isotropic Heisenberg models on amorphous lattices.

Overall, there has been surprisingly little research on amorphous quantum many body phases albeit material candidates aplenty. We expect our exact chiral amorphous spin liquid to find many generalisation to realistic amorphous quantum magnets and beyond.

## Data availability
The data used to produce these plots are of limited availability, due to the ease with which they can be generated from the publicly available code and, in the case of the evidence for the ground state flux sector, the large file size. Access can be obtained by contacting the authors.

## Code availability
The source code used to generate these results is publicly available online[98].

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

## Acknowledgements

We thank Adolfo Grushin and Cecille Repellin for helpful discussions and collaboration on related work. J.K. acknowledges support via the Imperial-TUM flagship partnership. The research is part of the Munich Quantum Valley, which is supported by the Bavarian state government with funds from the Hightech Agenda Bayern Plus. This work was supported in part by the Engineering and Physical Sciences Research Council (EP/T51780X/1 G.C., EP/R513052/1 T.H., P.D.).

## Author contributions

All authors contributed to the execution and write-up of this project. The numerical calculations were performed by G.C., P.D. and T.H. with additional contributions and analytical results by W.N. The project was initiated and supervised by J.K.

## Funding

## Competing interests
The authors declare no competing interests.
