## [Peer Review File · Nature Communications]

REVIEWER COMMENTS

Reviewer #1 (Remarks to the Author):

The authors of the present paper extended the Kitaev honeycomb model to amorphous lattices generated via Voronoi tessellation with fixed coordination number three. Based on this model, they investigated the role of spin-anisotropic interactions and quantum effects in amorphous magnets, and constructed a soluble chiral amorphous quantum spin liquid. They found that the presence of plaquettes with an odd-number of sites lead to a chiral quantum spin liquid with spontaneously broken time-reversal symmetry. The phase diagram of the model was presented, and it was shown that the chiral phase around the symmetric point is gapped and characterized by a quantized local Chern number as well as protected chiral Majorana edge modes. They also found that the amorphous system undergoes an Anderson transition to a thermal metal phase, driven by the proliferation of vortices with increasing temperature. In my opinion, this is an important and impressive work which has significant impact on the field of quantum spin liquid. This work paves the way to investigate topologically quantum many-body phase in amorphous magnets for future research. Therefore, I recommend publication in Nature Communications.

Reviewer #2 (Remarks to the Author):

The manuscript presents a generalisation of the Kitaev honeycomb model to 3-coordinated amorphous random graphs. It is argued using a combination of analytical and numerical tools, that the model remains exactly solvable, and realises a chiral spin liquid phase with spontaneous time-reversal breaking at low temperatures, thus demonstrating for the first time an inherently interacting amorphous topological phase. The authors also demonstrate that this phase is robust against bond-length dependence of the exchange couplings, and that at finite temperatures a transition into a thermal metal phase takes place. The manuscript is well written, the results are sound, and the results are timely, as there is significant interest in the community about amorphous topological phases of matter.

My only concern is about the experimental relevance of the chosen model Hamiltonian. Is the Hamiltonian local in the following sense: Can one expect a material sample to form, governed by short-range interactions, where the 3-coloring rules for the Ising directions is observed? Naively, the procedure for 3-coloring the edges appears to be highly non-local, can the authors propose any physical mechanism where such a low-energy Hamiltonian could arise in a solid with finite-range interactions?

Overall, I find the manuscript to be high quality and innovative, and I would recommend publication after the authors address my comment above.

Further minor comments:

- Typo in SM fig 4 caption “flux defect density, τ ” should be ρ .
- Typo on bottom of page 5 “but can take up to seven edges on the torus” should be seven colours.

Reviewer #3 (Remarks to the Author):

This is a well-written paper reporting a nontrivial and very interesting generalization of Kitaev’s honeycomb model to amorphous lattices. The main findings (summarized in the abstract and conclusion, need not to be repeated here) are novel, and its overall quality to my judgement is on par with some of the best physics papers I saw in Nature Communications. The work is expected to be stimulating to readers from quantum magnetism, topological physics, and disordered systems. My only complaint is that I have to go the supplementary materials often in order to appreciate how the main conclusions are reached, and what some of the symbols/terms mean. But I understand this perhaps has more to do with the format, style, and length constraints of the journal.

Comments:

1. Title. The word “exact” is better replaced by “exactly solvable” or something weaker. To my understanding, many of the results, including the ground state flux sector, the real space Chern index, the phase boundary, and finite-T behaviors, do rely on numerics on finite systems or involving averaging over random samples. It differs from any analytical exact solution in the thermodynamics limit. This (not being “exact”) of course does not impact the novelty of the main results.

2. The lattice generated by Voronoi tessellation. Given the central importance of “odd plaquettes” to the nature of the ground state, readers need to know the statistical composition of the plaquettes, e.g. what is the percentage of “odd” ones versus the “even” ones, and how the ratio fluctuates from sample to sample. I suspect the ratio has something to do with the inward curvature of the boundary of phase B.

3. The surprising and important result, Eq. (4), will be greatly clarified if the authors give an example (in Fig. 1b?) to illustrate the flux pattern in both phases. Unfortunately, the figure caption "Shading of the plaquettes indicates the Z2 flux $\phi_p = +1$ (-1) through even-sided white (grey) plaquettes $+i$, ($-i$) for odd sided plaquettes" is beyond my reading comprehension skills. A few questions are in order.

Is it true that one can only focus on the odd plaquettes to detect the breaking of TRS?

What is the key difference in the flux pattern for phase A and phase B? In the way it is written, Eq. (4) seems to be general, i.e. valid for all points on the phase diagram.

In phase B, does the gap correspond to flipping the flux of one of the odd plaquettes (or even ones as well)? This question is relevant because as one approaches the phase boundary, the gap will close, what is the mechanism behind the gap closing?

4. Figure 2. What exactly is being plotted in 2a? Here, does "density" or "wave function" refer to Majorana fermions? If so, its expression in terms of operator c (hopefully no b) would be nice. Similarly "fermionic density of states" or $\Psi(r)$ in the definition of IPR were not defined explicitly in the main text.

Moreover, what do these quantities represent physically? After all, the original model is for spins. For example, how to detect the edge states? It is chiral, do we expect some spin current? Spin-polarized STM and heat transport were mentioned in the end, but the connections between Figure 2 and these suggestions seem to be missing.

5. In several places, possible realization was mentioned. But it seems a little stretched to imagine a quenched magnetic material will realize the lattice considered (generated by Voronoi tessellation) with fixed $z=3$. Suppose some vertices or bonds are taken out or inserted due to uncontrolled sample preparation, would it destroy phase B?

Due to these additional disorder, would one also expect a spin glass region in the phase diagram, especially when the gap is small? Some argument would be illuminating.

I hope the authors can clarify some of these issues before I can recommend its publication.

Response to referee A:

The authors of the present paper extended the Kitaev honeycomb model to amorphous lattices generated via Voronoi tessellation with fixed coordination number three. Based on this model, they investigated the role of spin-anisotropic interactions and quantum effects in amorphous magnets, and constructed a soluble chiral amorphous quantum spin liquid. They found that the presence of plaquettes with an odd-number of sites lead to a chiral quantum spin liquid with spontaneously broken time-reversal symmetry. The phase diagram of the model was presented, and it was shown that the chiral phase around the symmetric point is gapped and characterized by a quantized local Chern number as well as protected chiral Majorana edge modes. They also found that the amorphous system undergoes an Anderson transition to a thermal metal phase, driven by the proliferation of vortices with increasing temperature. In my opinion, this is an important and impressive work which has significant impact on the field of quantum spin liquid. This work paves the way to investigate topologically quantum many-body phase in amorphous magnets for future research. Therefore, I recommend publication in Nature Communications.

We thank the referee for the concise summary and positive assessment of our work.

Response to referee B:

The manuscript presents a generalisation of the Kitaev honeycomb model to 3-coordinated amorphous random graphs. It is argued using a combination of analytical and numerical tools, that the model remains exactly solvable, and realises a chiral spin liquid phase with spontaneous time-reversal breaking at low temperatures, thus demonstrating for the first time an inherently interacting amorphous topological phase. The authors also demonstrate that this phase is robust against bond-length dependence of the exchange couplings, and that at finite temperatures a transition into a thermal metal phase takes place. The manuscript is well written, the results are sound, and the results are timely, as there is significant interest in the community about amorphous topological phases of matter.

My only concern is about the experimental relevance of the chosen model Hamiltonian. Is the Hamiltonian local in the following sense: Can one expect a material sample to form, governed by short-range interactions, where the 3-coloring rules for the Ising directions is observed? Naively, the procedure for 3-coloring the edges appears to be highly non-local, can the authors propose any physical mechanism where such a low-energy Hamiltonian could arise in a solid with finite-range interactions?

We thank the referee for the important question. First and foremost our work is a proof of principle showing for the first time that stable quantum spin liquids (or more broadly speaking topologically ordered) phases can exist on amorphous lattices. However, starting from the existing Kitaev materials we expect that successively including pentagon and heptagon defects would still give locally anisotropic spin interactions, thus, potentially realising an amorphous Kitaev spin liquid. The reason is that the anisotropies are dictated by the local bonding angles which can still be rather homogeneous even in an amorphous lattice (meaning that the fixed coordination number constraint in a relaxed structure always leads to bond angles around 120 degrees). Now of course, any local distortion will also change the relative strength of the perturbing Heisenberg/Gamma interactions in any realistic setting. It will be a challenging but very interesting study for future research to investigate in how far a Kitaev spin liquid can appear in a realistic material like the amorphous Li_2IrO_3 reported in Lee & Park, *Sci. Rep.* 9, 13180 (2019). One could for example perform ab-initio calculations with supercells including pentagon+heptagon defects and extract the exchange constants as has been done in incommensurate heterostructure with RuCl_3 , see Biswas et al., *Phys. Rev. Lett.* 123, 237201 (2019).

In this context, we note that amorphousness can also help to destabilise competing ordered phases, for example a simple antiferromagnetic (or zigzag) pattern is frustrated on odd plaquettes, which could potentially help to stabilise the liquid phase even in the presence of increased Heisenberg interactions.

Finally, one class of materials where amorphous Kitaev systems can potentially be realised are metal organic frameworks, which have been shown to exist in amorphous forms and which can host Kitaev interactions as discussed at the end of our manuscript.

Overall, I find the manuscript to be high quality and innovative, and I would recommend publication after the authors address my comment above.

Further minor comments:

- Typo in SM fig 4 caption “flux defect density, τ ” should be ρ .
- Typo on bottom of page 5 “but can take up to seven edges on the torus” should be seven colours.

We thank the referee for carefully reading through the manuscript. These errors have been in fixed.

Response to referee C:

This is a well-written paper reporting a nontrivial and very interesting generalization of Kitaev's honeycomb model to amorphous lattices. The main findings (summarized in the abstract and conclusion, need not to be repeated here) are novel, and its overall quality to my judgement is on par with some of the best physics papers I saw in Nature Communications. The work is expected to be stimulating to readers from quantum magnetism, topological physics, and disordered systems. My only complaint is that I have to go the supplementary materials often in order to appreciate how the main conclusions are reached, and what some of the symbols/terms mean. But I understand this perhaps has more to do with the format, style, and length constraints of the journal.

Comments:

1. Title. The word “exact” is better replaced by “exactly solvable” or something weaker. To my understanding, many of the results, including the ground state flux sector, the real space Chern index, the phase boundary, and finite-T behaviours, do rely on numerics on finite systems or involving averaging over random samples. It differs from any analytical exact solution in the thermodynamics limit. This (not being “exact”) of course does not impact the novelty of the main results.

In this context, the word exact refers to the fact that the system can be rigorously shown to be in a spin liquid phase, e.g. the fractionalization of spins into fluxes and Majoranas is exact. Of course, the ansatz for the ground state flux sector was supported by numerical arguments, however it does not change the fact that no approximations were necessary for any of our arguments. We believe that the current title communicates this, and so would prefer not to change it. Furthermore, the title is directly inspired by the foundational work of Yao and Kivelson (Phys. Rev. Lett. 99, 247203) and, therefore, we prefer to keep it.

2. The lattice generated by Voronoi tessellation. Given the central importance of “odd plaquettes” to the nature of the ground state, readers need to know the statistical composition of the plaquettes, e.g. what is the percentage of “odd” ones versus the “even” ones, and how the ratio fluctuates from sample to sample. I suspect the ratio has something to do with the inward curvature of the boundary of phase B.

All the lattices studied in this work were generated using a set of uniformly distributed seed points, and so have a very close to 50:50 distribution of odd and even plaquettes, in practice the ratio fluctuates, however for any reasonably large lattice this fluctuation is negligible. (For example, a quick calculation over a sample of 100 random lattices with 1000 plaquettes each gave an average proportion of odd plaquettes of 0.503 with a standard deviation around 0.015)

It is an interesting question to ask how the properties of the system depend on the ratio of odd and even plaquettes. We would like to draw the referees attention to a follow up paper to our work by Grushin and Repellin (Phys. Rev. Lett. 130, 186702), in which they studied the effect of slowly increasing amorphicity. The latter smoothly increases the ratio of odd to even plaquettes, showing that the chiral fermionic gap depends linearly on the proportion of odd plaquettes.

We have added a footnote ([69]) to the paper stating that the considered lattices always have $\sim 50\%$ odd plaquettes.

3. The surprising and important result, Eq. (4), will be greatly clarified if the authors give an example (in Fig. 1b?) to illustrate the flux pattern in both phases. Unfortunately, the figure caption “Shading of the plaquettes indicates the Z_2 flux $\phi_p = +1$ (-1) through even-sided white (grey) plaquettes +i, (-i) for odd sided plaquettes” is beyond my reading comprehension skills. A few questions are in order.

We thank the referee for pointing this out, diagram 1 and the caption have been edited to make this clearer.

Is it true that one can only focus on the odd plaquettes to detect the breaking of TRS?

Yes, we do believe so. Chiral symmetry is spontaneously broken in the amorphous Kitaev model (as well as any regular lattice with odd plaquettes) by the fact that the conserved flux operators W_p have imaginary eigenvalues on any odd plaquette – meaning that TRS will necessarily be broken by any choice of flux sector.

What is the key difference in the flux pattern for phase A and phase B? In the way it is written, Eq. (4) seems to be general, i.e. valid for all points on the phase diagram.

We believe that the same choice of ground state flux sector applies equally to the A and B phase. We have verified this numerically across the majority of the triangular phase diagram, although demonstrating this becomes increasingly difficult the closer to the corners of the triangle one is, i.e. when one of the J parameters is large and the other two are very small. This is because the flux gap (i.e. the energy of the lowest excited flux sector) scales with J_{small}^4/J_{large}^3 , so it becomes numerically very difficult to distinguish between the various low energy flux sectors when very close to a corner.

In phase B, does the gap correspond to flipping the flux of one of the odd plaquettes (or even ones as well)? This question is relevant because as one approaches the phase boundary, the gap will close, what is the mechanism behind the gap closing?

By gap closing, we always refer to the fermionic gap – that is staying in the ground state flux sector, one must determine whether the corresponding non-interacting Majorana Hamiltonian (the matrix iA – eq. 2 in the text) is gapped or gapless at zero energy. We always stay in the same canonical ground state flux sector given by eqn. 4. Gap closing is necessary when crossing the interface between the A and B phase, since the hamiltonian iA has a different Chern number in the two regimes.

4. Figure 2. What exactly is being plotted in 2a? Here, does “density” or “wave function” refer to Majorana fermions? If so, its expression in terms of operator c (hopefully no b) would be nice. Similarly “fermionic density of states” or $\Psi(r)$ in the definition of IPR were not defined explicitly in the main text.

We thank the referee for noticing this omission in our explanation of the method for solving the Kitaev Hamiltonian. By density and wave function, we are referring to the individual eigenvalues of the non-interacting Majorana operator (eqn. 2). In practice an eigenstate of the full Hamiltonian corresponds to filling half of the available Majorana states in the non-interacting system. We have added a section detailing this briefly, however a thorough exposition can also be found in section 3 of Kitaev’s original paper on the Honeycomb model (citation [31] in our paper), and the process was identical in our case.

We have also added some clarification to the paper, stating that ‘fermionic’ properties always refer to properties of the free fermionic Hamiltonian (eq. 2), which includes the fermionic density of states and IPR.

Moreover, what do these quantities represent physically? After all, the original model is for spins. For example, how to detect the edge states? It is chiral, do we expect some spin current? Spin-polarized STM and heat transport were mentioned in the end, but the connections between Figure 2 and these suggestions seem to be missing.

We thank the referee for this question as it touches upon one of the most subtle questions - how to observe spin fractionalization in any realistic experiments. The difficulty arises because local spin flip excitations necessarily excite several Majoranas and fluxes and therefore, the chiral edge modes would not transport any spin current. However, they are physical excitations and therefore can

transport energy which can be measured as a thermal Hall signal. Alternatively, there are ideas how to detect these edge states via STM, see Feldmeier et al., Phys. Rev. B. 102, 134423 (2020).

5. In several places, possible realization was mentioned. But it seems a little stretched to imagine a quenched magnetic material will realize the lattice considered (generated by Voronoi tessellation) with fixed $z=3$. Suppose some vertices or bonds are taken out or inserted due to uncontrolled sample preparation, would it destroy phase B?

There are two different answers to this question. First, one could think of taking our amorphous lattices and remove sites, which has been extensively studied for the honeycomb case, see e.g. Willans et al., Phys. Rev. B 84, 115146. In this case the spin liquid is stable but additional local moments can emerge from flux binding on the defect sites. Second, one could ask whether any $z=3$ amorphous lattice is naturally formed. In fact, a fixed coordination number is quite natural and appears for example in amorphous graphene, see e.g. Fig.2 d of Toh et al., Nature 577, 199-203 (2020), and possibly even in the Kitaev material Li_2IrO_3 , see Lee & Park, Sci. Rep. 9, 13180 (2019). In general, since the phase B is a genuine topologically ordered phase it is robust w.r.t. to arbitrary perturbations (like additional interactions or disorder) as long as these are not too strong.

Due to these additional disorder, would one also expect a spin glass region in the phase diagram, especially when the gap is small? Some argument would be illuminating.

In the pure Kitaev limit we do not expect a spin glass to form, but in principle with the addition of Heisenberg terms one might see these effects. We expect that removing sites or bonds in our pure Kitaev limit will not destroy the fractionalised phase, but might lead to localisation of the fermionic states, which would correspond to a quantum spin glass forming. This is, however, an interesting question and a more thorough investigation of glassy physics in such a model would be very worthwhile in the future.

I hope the authors can clarify some of these issues before I can recommend its publication.

We thank the referee for the thoughtful questions. We hope that our responses could clarify these points and that our paper is now ready for publication.

REVIEWERS' COMMENTS

Reviewer #3 (Remarks to the Author):

The authors have satisfactorily addressed all my comments, and improved the manuscript. I recommend the publication of the revised version.